# Green Manure Rotation Combined with Biochar Application Improves Yield and Economic Stability of Continuous Cropping of Peppers in Southwest China

**DOI:** 10.3390/plants13233387

**Published:** 2024-12-02

**Authors:** Meng Zhang, Yanling Liu, Xiaofeng Gu, Quanquan Wei, Lingling Liu, Jiulan Gou

**Affiliations:** Institute of Soil and Fertilizer, Guizhou Academy of Agricultural Sciences, Guiyang 550006, China

**Keywords:** green manure plowing and returning to the field, wine lees biochar, obstacle soil, nutritional quality, production sustainability

## Abstract

Crop rotation is widely recognized as a key strategy to mitigate the adverse effects associated with continuous cropping. Recent studies have demonstrated that biochar has a significant potential for preventing and controlling these challenges. However, the ameliorative effects of green manure rotation and biochar application on continuous pepper cultivation in the karst mountainous regions of Southwest China remain largely unexplored. To address this gap, a field experiment was conducted from 2020 to 2023 to investigate the effects of green manure rotation and biochar application on the continuous cropping of peppers. The experiment consisted of five treatments: CK (no green manure and no biochar), WP (winter fallow and conventional pepper production with chemical fertilization), GP (green manure and pepper rotation, the amount of fresh green manure returned to the field was about 15 t·ha^−1^), WP + B (winter fallow and pepper rotation with 1500 kg·ha^−1^ of biochar applied during the pepper season), and GP + B (green manure and pepper rotation with 1500 kg·ha^−1^ of biochar applied during the pepper season, the amount of fresh green manure returned to the field was about 15 t·ha^−1^). The results showed that all the improved measures (GP, WP + B, GP + B) increased the yield of fresh pepper and dry pepper by 26.97–72.98% and 20.96–65.70%, respectively, and the yield of dry pod pepper increased by 14.69–40.63% and 21.44–73.29% in 2021 to 2023, respectively, and significantly improved the yield stability and sustainability of continuous cropping of peppers compared with WP treatments. In addition, green manure rotation or biochar application alone or in combination enhanced the nutritional quality of pepper fruits by increasing the content of free amino acids (8.62–19.42%), reducing sugars (15.30–34.62%) and vitamin C (26.19–43.52), and decreasing the nitrate content (26.93–40.17%). Furthermore, the application of green manure rotation or biochar alone or in combination significantly improved the absorption of nitrogen (23.73–60.23%), phosphorus (18.12–61.71%), and potassium (20.57–61.48%) nutrients in the continuous cropping of peppers, which contributed to the improvement of fertilizer use efficiency. Notably, GP + B treatment not only improved the yield and quality of continuous cropping peppers but also resulted in higher production value and net income compared to the GP and WP + B treatments. In conclusion, the combination of green manure rotation and biochar application represents an effective strategy for mitigating the challenges of continuous cropping in pepper cultivation within the karst mountainous regions of Southwest China.

## 1. Introduction

The demand for food and cash crops has increased with the ongoing growth of the world’s population. Due to the limited availability of arable land and the decreasing amount of land available for new cultivation, continuous cropping has emerged as the predominant cropping pattern in intensive agricultural production [1]. Despite the implementation of effective management practices, crops cultivated under prolonged continuous cropping systems are prone to experiencing continuous cropping disorders. These disorders manifest as slowed growth, decreased productivity, lower crop quality, and an increased incidence of soil-borne diseases [2]. Prolonged continuous cropping results in several complications, such as the degradation of soil physicochemical conditions, reduced soil enzyme activity, accumulation of harmful substances, and disruptions in soil microbial populations [3].

In modern monoculture cultivation models, the growth and reproduction rates of pathogenic microorganisms during crop development are significantly higher than those observed with traditional crop rotation practices [4,5]. During crop growth, root secretion from plants can increase the population of pathogenic microorganisms [6,7]. Moreover, with an increase in continuous cropping years, beneficial microorganisms in the soil gradually decrease, whereas pathogenic microorganisms accumulate in large quantities, which ultimately reduces the self-healing ability of the soil, leading to an imbalance in soil microecology and an increase in the incidence rate of soil-borne diseases [8,9]. Crop rotation is a crucial strategy for effectively addressing challenges associated with continuous cropping. This practice primarily involves diversifying the variety of crops within the cropping system to reduce continuous cultivation of the same crop [10,11]. Some studies have found that crop rotation can effectively reduce the soil fungal population, increase the ratio of soil bacterial to fungal content (B/F), improve the microbial diversity index, ameliorate soil physicochemical properties and microbiota, and reduce plant morbidity and mortality [12,13]. In addition, a well-planned crop rotation pattern not only balances the soil microbial community but also improves the root vitality of subsequent crops [14]. However, it is worth noting that due to low winter temperatures and shorter hours of sunshine in the karstic mountains, most crops are unable to adapt to this environment and grow normally or slowly. In recent years, planting green manure during the winter season has shown to be a valuable strategy for addressing the challenges posed by continuous cropping. Pál and Zsombik [15] found that green manure rotations can be good for increasing maize yields, especially under drought stress conditions. Liu et al. [16] reported that green manure improves soil quality and ecosystem multifunctionality by increasing soil available nutrients and extracellular enzyme activities under the paddy system. Many studies have confirmed that green manure rotation not only enhances crop yield and quality but also improves the soil microbial environment. Consequently, it has emerged as a vital technical approach to achieving sustainable agricultural development [17,18].

Exogenous organic materials such as biochar and bioorganic fertilizers have received significant attention in recent years because of their effectiveness in enhancing soil quality [19,20]. Exogenous organic materials can significantly reduce the incidence of soil-borne diseases by suppressing pathogenic bacteria, enhancing nutrient levels, and neutralizing the soil pH [21]. In recent years, exogenous organic materials have emerged as a popular research topic for mitigating and managing the challenges associated with continuous cropping. Biochar has been found to mitigate the negative effects of soil autotoxicity in continuous cropping systems because of its high specific surface area, porosity, and adsorption capacity [22]. Moreover, biochar creates an excellent habitat for soil microorganisms, promotes the growth of beneficial microorganisms, and maintains microbial activity [23]. Studies have demonstrated that biochar can not only adsorb harmful substances in the soil and reduce the content of self-toxic substances in the field but can also mitigate the effects of these toxic substances on plant growth. In addition, biochar enhances the biomass and growth rate of beneficial microorganisms [24,25]. Some studies have reported that biochar can help plant roots effectively attract disease-suppressive inter-root microbiomes for disease control by enhancing the chemotaxis of beneficial microorganisms through root exudates and promoting biofilm formation [26]. A recent study has confirmed that the application of biochar significantly increases the sugar content of sugar beets and significantly reduces molasses content, and that biochar application is effective in mitigating the adverse environmental impacts of global warming potential (GWP), freshwater eutrophication (FEP), and ocean eutrophication [27]. Therefore, biochar may have significant potential as a soil amendment material to address the issues of continuous cropping.

Chili peppers (*Capsicum annuum* L.) are rich in nutrients, including vitamin C (VC) and capsaicin, and have significant nutritional and economic value [28,29]. The cultivated area of peppers in China is >2.1 million hectares, with >50% cultivated in Southwest China. However, owing to inadequate cultivation and management practices, continuous cropping of peppers has become a prevalent issue in the pepper-growing regions of China. Continuous cropping obstacles, such as soil acidification, soil nutrient imbalance, and the increased prevalence of pests and diseases, have become increasingly prominent and have seriously limited the sustainable development of the pepper industry in China [30]. It is important to note that the challenges associated with continuous pepper cropping are particularly serious in the mountainous regions of southwest China. This challenge is primarily attributed to the limited availability of arable land resources and excessive reliance on chemical fertilizers. While various field studies have demonstrated the effectiveness of green manure rotation and biochar in alleviating continuous cropping issues, their specific impacts on pepper cultivation in the karst mountainous regions remain underexplored. In this research, we hypothesized that green manure return and biochar application alone or in combination would have an ameliorative effect on continues cropping peppers; therefore, we conducted a three-year field trial with the following research objectives: (1) to assess the effects of green manure rotation or biochar application on yield stability and sustainability of continuous cropping peppers; (2) to investigate the effects of green manure rotation or biochar application alone or in combination on nutrient accumulation and fertilizer utilization efficiency; and (3) to evaluate the effects of green manure rotation or biochar application on production value and net income of continuous cropping peppers. The results of this research provide valuable insights into the promotion of sustainable pepper cultivation in karst mountainous regions.

## 2. Results

### 2.1. Yield of Continuous Cropping Peppers

The fresh pepper yields of the WP treatments gradually decreased as the number of years in which the peppers were planted increased (Figure 1A). The WP treatment resulted in fresh pepper yields that were 8.64% lower in 2022 and 10.52% lower in 2023 than those in 2021. In contrast, green manure return or combined with biochar application (GP, WP + B, and GP + B), led to an increase in fresh pepper yields ranging from 26.97% to 72.98% on average compared to the WP treatment. Interestingly, compared with the GP and WP + B treatments, the fresh pepper yield for the GP + B treatment increased by an average of 25.35% and 36.68%, respectively. In terms of dry pepper yield (Figure 1B), the application of green manure, either alone or in combination with biochar (GP, WP + B, and GP + B), resulted in an average increase in dry pepper yield between 20.96 and 65.70% compared with WP treatment. In addition, the results of the study similarly showed that the yield of dry pepper increased by an average of 25.58% in the GP + B treatment compared with the GP treatment and by 37.72% compared with the WP + B treatment.

### 2.2. Yield Stability and Sustainability of Continuous Cropping Peppers

The introduction of green manure and biochar notably improved the yield stability and sustainability of continuous pepper cropping. Compared to the WP treatment, the CV for fresh pepper in green manure return or combined with biochar application treatments (GP, WP + B, and GP + B) decreased by 7.09–8.44%, whereas the SYI increased by 9.36–11.95% (Figure 2A,C). Compared with the GP and WP + B treatments, the CV of fresh pepper in the GP + B treatment decreased by 0.22% and 1.35%, respectively, whereas the SYI increased by 0.79 and 2.59%, respectively. The findings also highlighted that the CV of dry peppers in the green manure return or combined with biochar application treatments (GP, WP + B, and GP + B) decreased by 2.91–4.49% compared with the WP treatment, whereas the SYI increased by 5.42–8.05% (Figure 2B,D). Similarly, compared with the GP and WP + B treatments, the CV of dry peppers in the GP + B treatment decreased by 0.05 and 1.58%, respectively, whereas the SYI increased by 0.50 and 2.63%, respectively.

### 2.3. Nutritional Quality of Fresh Peppers Fruits

The use of green manure and biochar significantly enhanced the nutritional quality of pepper fruits (Table 1). When compared to the WP treatment, the average contents of free amino acids, reducing sugars, and vitamin C (VC) in the green manure return or combined with biochar application treatments (GP, WP + B, and GP + B) increased by 8.62–19.42%, 15.30–34.62%, and 26.19–43.52%, respectively. From 2021 to 2023, the GP + B treatment further elevated the average levels of free amino acids, reducing sugars, and VC by 7.99%, 13.10%, and 8.24%, respectively, compared to the GP treatment, and by 9.58%, 16.81%, and 14.26%, respectively, compared to the WP + B treatment. Moreover, in comparison to the WP treatment, the green manure and biochar applications (GP, WP + B, and GP + B) decreased the average nitrate content in fresh pepper fruits between 26.93% and 40.17% over the three-year period. In addition, the average nitrate content in the GP + B treatment decreased by 16.98% and 20.28% compared with the GP and WP + B treatments, respectively.

### 2.4. Nutrient Accumulation in Continuous Cropping Peppers

We measured the NPK nutrient content of pepper plants (including stems, leaves, and pepper fruits) and ultimately calculated the NPK nutrient accumulation of the entire pepper plant. Figure 3 illustrates that the application of green manure return and biochar significantly affected the accumulation of NPK nutrients. As the number of pepper cultivation years increases, the NPK nutrient accumulation in WP treatments did not increase significantly, but on the contrary, showed a decrease in some years. Compared with 2021, N accumulation in WP treatments increased only 8.17% in 2023, while P and K accumulation decreased by 3.34% and 2.98%, respectively. Compared to the WP treatment, the average accumulation of N, P, and K nutrients in treatments involving green manure return or combined with biochar application (GP, WP + B, and GP + B) increased by 23.73–60.23%, 18.12–61.71%, and 20.57–61.48%, respectively. Interestingly, the average accumulation of N, P, and K in the GP + B treatment increased by 21.21%, 22.10%, and 21.25% from 2021 to 2023 compared to the GP treatment, whereas they elevated by 29.71%, 36.65%, and 34.20% in the GP + B treatment compared with the WP + B treatment.

### 2.5. Fertilizer Utilization in Continuous Cropping Peppers

The use of green manure return and biochar significantly increased fertilizer efficiency in the continuous cropping of peppers (Table 2). Compared with the WP treatment, the average AE_N_, AE_P_, and AE_K_ of the green manure return or the combined biochar application treatments (GP, WP + B, and GP + B) increased by 1.83–6.11 kg·kg^−1^, 6.93–23.07 kg·kg^−1^, and 2.15–7.14 kg·kg^−1^, respectively. The results also showed that the average AE_N_, AE_P_, and AE_K_ of the GP + B treatment increased by 3.18 kg·kg^−1^, 12.03 kg·kg^−1^, and 3.72 kg·kg^−1^, respectively, compared to the GP treatment, whereas they elevated by 4.27 kg·kg^−1^, 16.14 kg·kg^−1^, and 5.00 kg·kg^−1^ in GP + B treatment compared with WP + B treatment. In terms of RE, the green manure return or combined with biochar application treatments (GP, WP + B, and GP + B) increased RE_N_, RE_P_, and RE_K_ by an average of 8.98–22.74, 5.38–18.41, and 11.82–35.48%, respectively, compared with the WP treatments. Notably, RE_N_, RE_P_, and RE_K_ in the GP + B treatment increased by an average of 10.39, 8.82%, and 16.27%, respectively, compared with the GP treatment, whereas RE_N_, RE_P,_ and RE_K_ in the GP + B treatment increased by an average of 13.77, 13.03, and 23.66%, respectively, compared with the WP + B treatment.

### 2.6. Economic Benefits of Continuous Cropping Peppers

Figure 4 illustrates that the use of green manure return and biochar significantly improved the economic returns of continuous pepper cropping. Compared with the WP treatment (Figure 4A), the average output value of dry peppers with green manure return or combined with biochar (GP, WP + B, and GP + B) increased by 20.96–65.70%. The output value in the GP + B treatment increased by 27.23%, 27.79%, and 21.72% from 2021 to 2023, respectively, compared with the GP treatment. In contrast, the GP + B treatment showed increases of 41.36%, 41.07%, and 30.72% compared with the WP + B treatment. In terms of net income from dry peppers (Figure 4B), the average net income of dry peppers in green manure return or the combined application of biochar (GP, WP + B, and GP + B) increased by 23.80–66.57%. In addition, the results indicated that the net income from dry peppers in the GP + B treatment increased by an average of 22.39% and 35.43% compared with the GP and WP + B treatments, respectively.

## 3. Discussion

### 3.1. Green Manure Rotation and Biochar Application Improve Yield and Quality of Continuous Cropping Peppers

The long-term practice of continuous cropping has become a widespread trend in modern agriculture [3]. With an increase in the number of years of continuous cropping, crop growth has been affected to varying degrees [31,32]. Challenges associated with continuous cropping have gradually emerged as important barriers to the sustainable development of green agriculture. The results of this study found that the green manure–pepper rotation (GP treatment) increased fresh and dry pepper yield by 38.21% and 32.32%, respectively, while biochar application (WP+B) improved fresh and dry pepper yield by 26.97% and 20.96%, respectively, compared to WP treatment. This implies that planting green manure or increasing the application of biochar can be an effective solution to the problem of reduced yields caused by crop barriers. Previous studies have indicated that both green manure and biochar application can directly increase the biodiversity of farmland ecosystems by providing more habitat for soil fauna and microorganisms [33,34]. This increase in biodiversity subsequently enhances the composition of the food chain and the population of natural enemies, thereby reducing the incidence of pests and diseases, which is crucial for enhancing crop yield [35,36].

Improving the quality of agricultural products is the core of high-quality agricultural development. The results of this study also found that green manure rotation or application of biochar increased free amino acid (12.75%), reducing sugar (22.98%), and VC (34.15%) content in pepper fruits and significantly reduced nitrate (32.26%) content. It was found that because of the stress experienced by chili pepper plants from soil habitat degradation under continuous cropping conditions, chlorophyll synthesis was inhibited by reduced chlorophyll synthase activity, whereas the rate of chlorophyll degradation increased, resulting in a reduced rate of photosynthesis [37,38,39]. Furthermore, prolonged continuous cropping leads to a reduction in the activity of key antioxidant enzymes, such as POD and SOD, in the leaves, which compromises the plant’s immune defenses [40]. This decline in enzyme activity is detrimental to the stress tolerance of plants and adversely affects the nutritional quality of the fruits [38]. There is no doubt that the application of green manure or biochar significantly ameliorated this negative effect. This may be attributed to the increase in overall crop resistance due to the increase in antioxidant enzyme activity and the accumulation of osmoregulatory substances in the plant [19,41]. On the other hand, studies have also shown that the improvement of quality through biochar or green manure rotation may have a more or less synergistic effect with the improvement of soil fertility (or basic soil health) and plant/root growth [42].

### 3.2. Green Manure Rotation and Biochar Application Improve Nutrients Accumulation and Quality of Continuous Cropping Peppers

Reduced nutrient accumulation and fertilizer use efficiency are other prominent problems of continuous crop barriers. In the present study, it was found that NPK accumulation was increased by 32.54%, 32.18%, and 33.35% in GP treatment as compared to WP treatment, while agronomic efficiency (AE) and apparent utilization (RE) were also significantly enhanced in GP treatment. This improvement is likely due to the ability of leguminous green manure to facilitate nitrogen fixation through rhizobacteria, leading to the continuous accumulation of nitrogen in the soil after green manure was incorporated [43,44,45]. This process is beneficial for improving soil nitrogen content and utilization efficiency. However, the roots of leguminous green manure can secrete significant amounts of acids, enzymes, and other substances that are beneficial for activating phosphorus and potassium, which are often difficult to utilize in soil [46,47]. Similarly, NPK accumulation increased by 23.73%, 18.12%, and 20.57% in WP+B treatment compared to WP treatment, respectively. This positive effect of biochar is mainly due to the fact that biochar application notably increases the abundance of bacteria in continuous cropping soils, leading to greater stability within the soil microbial communities [48,49]. Moreover, the application of biochar not only enhances the population of beneficial microbes but also promotes the secretion of antibiotics, thereby reducing the prevalence of pathogenic bacteria [50].

However, it is important to note that the improvement effect of the WP + B treatment on the continuous cropping of peppers was not as good as that of the GP treatment in terms of nutrient accumulation, and fertilizer utilization efficiency in this study. This discrepancy could be linked to the prolonged use of biochar, which may reduce soil organic carbon mineralization, slow down organic carbon decomposition, and result in excessive carbon accumulation in the soil. This results in an imbalance in the carbon-to-nitrogen (C/N) ratio, which may limit soil microbial reproduction and hinder crop growth to some extent [51,52]. In addition, many studies have found that biochar produces many organic pollutants (such as volatile organic compounds (VOCs), polycyclic aromatic hydrocarbons (PAHs), and dioxins) during pyrolysis, and heavy metals in the raw materials can also accumulate in biochar, which may have a negative impact on soil environment, microorganisms, and even plant growth and development [21,53,54]. Therefore, it is necessary to assess the safety of biochar or feedstock before biochar is applied as a soil remediation material so as to minimize negative impacts at a later stage.

### 3.3. Green Manure Rotation Combined with Biochar Application Enhances the Production Capacity of Continuous Cropping Peppers

Notably, the results of this study also showed a decrease in CV by 0.14 and 5.83 percentage points, while SYI increased by 0.65 and 2.61 percentage points in GP+B treatment compared to GP and WP+B treatments, respectively. This means that the GP+B treatment showed greater yield stability and sustainability than the GP and WP+B treatments. In addition, GP+B treatment significantly improved pepper fruit quality, nutrient accumulation, fertilizer use efficiency, and even economic benefits. This may be due to the fact that the combination of green manure rotation and biochar application is more effective in alleviating the challenges associated with the continuous cropping of peppers. Studies have demonstrated that the combined application of biochar and green manure not only promoted the degradation of root secretions and plant residues by increasing extracellular enzyme activity but also reduced the population of saprophytic fungi and harmful secondary metabolites that negatively affect plant growth, and the reduction may contribute to lowering crop morbidity rates and improving soil health [55,56]. Green manure can attract beneficial microorganisms that degrade lipid chemosensory substances by increasing the accumulation of specific sugar metabolites in the soil [57,58]. This process can help alleviate the adverse effects of continuous cropping on yield and quality, and the ameliorative effect can become stronger with the synergistic application of biochar [59]. It is worth noting that parameters such as microbial biomass and diversity, soil porosity, bulk weight, water holding capacity, cation exchange capacity, pH value, etc., were changed after adding biochar to the soil combined with green manure, and such changes were related to the biochar species [60]. Therefore, the study of biochar species should be expanded in future research to obtain more accurate results.

## 4. Materials and Methods

### 4.1. Site Description

The field experiment was carried out from 2021 to 2023 in Zhongxin Village, Xinmin Town, Bozhou District, Zunyi City, Guizhou Province, China (27°22′58″ N, 106°56′15″ E). This region has a subtropical monsoon climate, with an altitude of 845 m, an annual sunshine of 1137 h, an average annual rainfall of 1100 mm, an average annual temperature of 15.3 °C, and an average frost-free period of 263 days. The test area serves as an important production hub for peppers in southwest China, where peppers are an important cash crop that contributes to local agricultural production and farmers’ incomes. Before this field trial, the test area had been under continuous pepper cultivation from 2018 to 2020. The planting pattern involved growing peppers during summer and leaving them fallow in winter.

The soil at the site is classified as yellow soil, a predominant type in this region. The yellow soil type at this region was classified as Acrisol in the World Reference Base for Soil Resources (WRB), and was developed from the Triassic limestone and sand shale efflorescence. Due to intense leaching caused by the perennial humidity, the exchangeable base content was only 20%. In addition, the yellow soil of the area is clay-heavy with a clay grain content of 581 g·kg^−1^. Soil samples were collected for testing and analysis after peppers were harvested in 2020. The primary physical and chemical characteristics of the soil were determined as follows: pH of 5.94, SOM (soil organic matter) content of 17.53 g·kg^−1^, TN (total nitrogen) content of 1.19 g·kg^−1^, TP (total phosphorus) content of 0.18 g·kg^−1^, AP (available phosphorus) content of 11.85 mg·kg^−1^, TK (total potassium) content of 25.34 g·kg^−1^, AK (available potassium) content of 172.97 mg·kg^−1^, and CEC (cation exchange capacity) content of 13.9 cmol·kg^−1^.

### 4.2. Experimental Material

The pepper variety selected for this experiment was “Zunla No. 9”. Currently, the planting area for this variety in the test area is >70%, making it the predominant pepper variety in the local area. The green manure used in the experiment was common vetch (*Vicia sativa* L.). Common vetch is the main winter green manure species variety planted locally, which is suitable for the local climate environment and has the characteristics of large biomass and abundant nutrients. Compound fertilizers CF1 (N-P_2_O_5_-K_2_O = 15-14-16, Guizhou Tianbao Fengyuan Ecological Agricultural Technology Co., Ltd., Xiuwen, China) and CF2 (N-P_2_O_5_-K_2_O = 18-6-18, Guizhou Tianbao Fengyuan Ecological Agricultural Technology Co., Ltd., Xiuwen, China) were used during pepper cultivation, where CF1 was used as a base fertilizer and CF2 was used as a dressing fertilizer. The biochar used in this study was prepared by carbonizing distillers’ grains at 550 °C using the biomass carbonization furnace (SSDP-5000-A, Jiangsu Huaian Huadian Environmental Protection Machinery Manufacturing Co., Ltd., Huaian, China) at the experimental base of Guizhou Academy of Agricultural Sciences. Biochar is in powder form and has an excellent pore structure. The physico-chemical properties of test biochar were as follows: pH of 8.78, organic carbon content of 342.71 g·kg^−1^, TN content of 15.63 g·kg^−1^, TP content of 1.05 g·kg^−1^, TK content of 19.78 g·kg^−1^, specific surface area (SSA) of 2.12 m^2^·g^−1^, single point adsorption total pore volume (SPATPV) of 2.95 × 10^−3^ cm^3^·g^−1^, and average pore size (APS) of 5.55 nm.

### 4.3. Experimental Design

The field experiment was carried out in September 2020 after the peppers were harvested. Subsequently, the trial plots were tilled with large machinery to ensure relatively consistent base fertility for the trial. The experiment consisted of five treatments; each replicated three times: (1) CK: a null check without fertilization; (2) WP: a conventional check, conventional pepper production with chemical fertilization; (3) GP: a green manure treatment, green manure, and pepper rotation; (4) WP + B: a biochar treatment, add biochar on the basis of WP treatment; (5) GP + B: a combined treatment of green manure and biochar. The planting pattern was consistent with that of the GP treatment, and biochar was applied during the pepper growing season. A total of 75 kg·ha^−1^ of green manure (common vetch) was sown in mid-October each year and incorporated into the soil using a rototiller in mid-March the following year (at full bloom). No chemical fertilizer was applied to any of the treatments during the green manure season. During the annual pepper season, CF1 and CF2 fertilizers were applied to all treatments except for the CK group, which did not receive any fertilizer. The application rate of CF1 and CF2 was 720 kg·ha^−1^, respectively. In addition, 1500 kg·ha^−1^ of biochar was applied annually in the WP + B and GP + B treatments, along with the application of CF1 and CF2. Fifteen days before transplanting peppers, CF1 or biochar were applied to the soil surface, mixed well with 0–30 cm of soil using a rotary tiller, and subsequently covered with mulch. After 15 days, the peppers were transplanted at a density of 45,000 plants·ha^−1^. Each treatment plot covered an area of 45 m^2^ (10.0 m × 4.5 m). CF2 was applied at the peak flowering stage of peppers. Appropriate pest and disease control measures were implemented as required during the growth of the green manure and peppers. Specific details of each treatment are shown in Table 3.

### 4.4. Sampling and Analysis

Before planting the green manure in October 2020, soil samples (0–20 cm depth) were collected from 15 randomly chosen locations across the experimental plots. These samples were subsequently transported to the laboratory, where debris, including plant and animal residues, was removed, naturally dried, and pulverized for testing and analysis. The pH, SOM, TN, TP, AP, TK, AK, and CEC contents were determined according to the method of Bao [61]. Six pepper plants were collected from each experimental plot during the annual pepper harvest and transported to the laboratory. Plant samples were prepared according to previously described methods, and TN, TP, and TK content were determined [61]. In addition, 2 kg of fresh pepper fruit was collected from each experimental plot during the pepper harvesting period to measure the levels of free amino acids, reducing sugars, vitamin C (VC), and nitrate content. Free amino acid content was determined using the ninhydrin colorimetric method. Reducing sugar content was determined using the 3,5-dinitrosalicylic acid method. The vitamin C (VC) content was determined using an HPLC (high-performance liquid chromatography, LC-2040, Nexera-i, Shimadzu, Tokyo, Japan) after grinding, centrifuging and filtering with 10 mL 0.2% metaphos phoric acid. Nitrate content was determined using a visible spectro photometer (UV-3600i Plus, Shimadzu, Tokyo, Japan).

It should be noted that the pepper yield was calculated individually based on the ripening and harvesting of peppers in each experimental plot because of the inconsistent ripening time of the pepper fruits. The specific steps were as follows. Mature fresh peppers from each test plot were harvested individually and were measured for fresh pepper yield. Fresh peppers from each test plot were then sent to a drying facility for drying and finally, the yield of dry pepper was measured. The ultimate yield of fresh and dry peppers was ascertained by gauging the combined weight across multiple harvests.

### 4.5. Calculations and Statistical Analysis

#### 4.5.1. Yield Stability

Yield stability was expressed utilizing the coefficient of variation (CV), which was used to measure the degree of variability among average pepper yields between years. Smaller values of CV indicate higher yield stability. The calculation method for yield stability is as follows [62]:$$CV=\frac{\sigma }{Y_{A}}\times 100\%$$where σ stands for standard deviation of yield (kg·ha^−1^), Y_A_ denotes the average yield (kg·ha^−1^).

#### 4.5.2. Yield Sustainability

Yield sustainability was expressed using the yield sustainability index (SYI). Higher values of SYI indicate higher sustainability of the system. The calculation method for yield sustainability is as follows [63]:$$SYI=\frac{Y_{A}–\sigma }{Y_{MAX}}$$
where σ stands for standard deviation of yield (kg·ha^−1^), Y_A_ represents the average yield (kg·ha^−1^), and Y_MAX_ denotes the maximum value of yield in all years (kg·ha^−1^).

#### 4.5.3. Fertilizer Utilization

Agronomic efficiency (AE) indicates the contribution of fertilizer to the increase in crop yield. Recovery efficiency (RE) indicates the efficiency of fertilizer uptake and utilization by crop. The calculation method for AE and RE are described in [64,65].

#### 4.5.4. Economic Benefits

The output value (OV) indicates the sales value of dry peppers per unit area. Net income (NEI) indicates profit after deducting production costs. The calculation method for OV and NEI are as follows:OV=Y×UPNEI=OV−FV
where OV stands for the output value (yuan·ha^−1^), Y represents the yield of dry pod pepper (kg·ha^−1^), UP denotes the unit price of dry pod pepper (yuan·kg^−1^), NEI signifies the net income (yuan·ha^−1^), and FV indicates the fertilizer value (yuan·ha^−1^). In assessing economic benefits, the unit price of dried pepper was 20.00 yuan·kg^−1^. The unit prices of CF1, CF2 and biochar was 3350, 3500 and 1000 yuan·t^−1^, respectively.

### 4.6. Statistical Analysis

All experimental data were analyzed utilizing Microsoft Excel 2019. Statistically significant differences between soil groups were estimated utilizing one-way ANOVA (*p* < 0.05; LSD and Duncan test) with Statistical Product and Service Solutions Statistics software (SPSS 23; IBM).

## 5. Conclusions

This study demonstrated that both green manure rotation and biochar application improved the yield of continuous cropping of peppers. Additionally, these practices have contributed to the stability and sustainability of pepper yield over time. Green manure rotation and biochar application also enhanced nutrient accumulation and increased fertilizer utilization efficiency. Overall, the combination of green manure rotation with biochar application proved to be more effective than either practice alone in improving the yield of continuous cropping of pepper. Therefore, it is recommended that a combined approach of green manure rotation and biochar application be implemented to address the challenges of continuous pepper cropping in karst mountainous regions. This strategy will help maintain and enhance the production capacity of peppers in the region. However, this study only evaluated the effectiveness of small areas in China and did not conduct in-depth research on soil microecology, which has certain limitations. In the future, we will establish a monitoring site for long-term localized experiments, and future research will focus on investigating the interactions between soil microorganisms, root secretions, and soil health.

## Figures and Tables

**Figure 1 plants-13-03387-f001:**
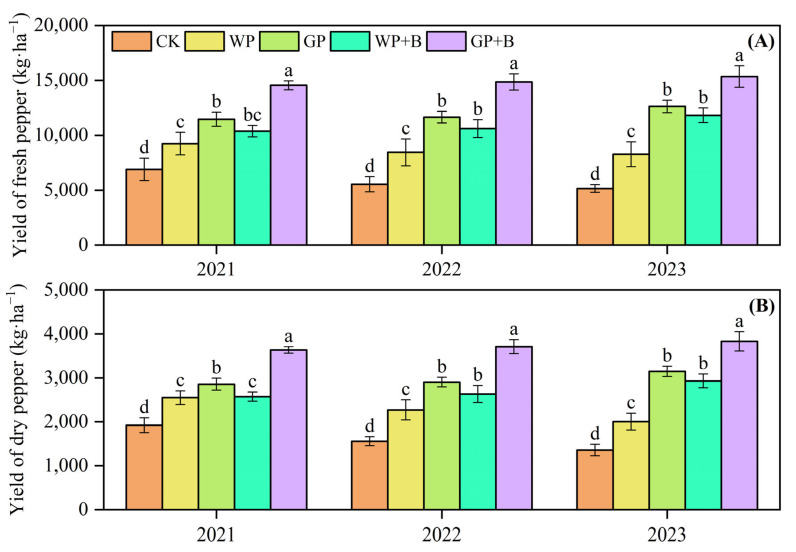
The effects of green manure return and biochar application on fresh (**A**) and dry (**B**) yield of continuous cropping pepper. Different lowercase letters indicate significant differences among different treatments at *p* < 0.05 by the Duncan’s MRT method.

**Figure 2 plants-13-03387-f002:**
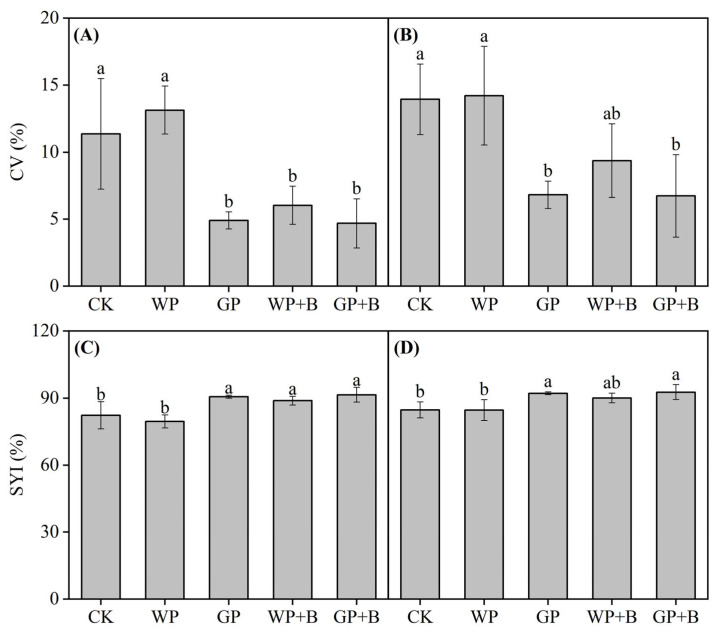
The effects of green manure return and biochar application on yield stability (**A**) and sustainability (**C**) of fresh peppers, and yield stability (**B**) and sustainability (**D**) of dry peppers. Different lowercase letters indicate significant differences among different treatments at *p* < 0.05 by the Duncan’s MRT method.

**Figure 3 plants-13-03387-f003:**
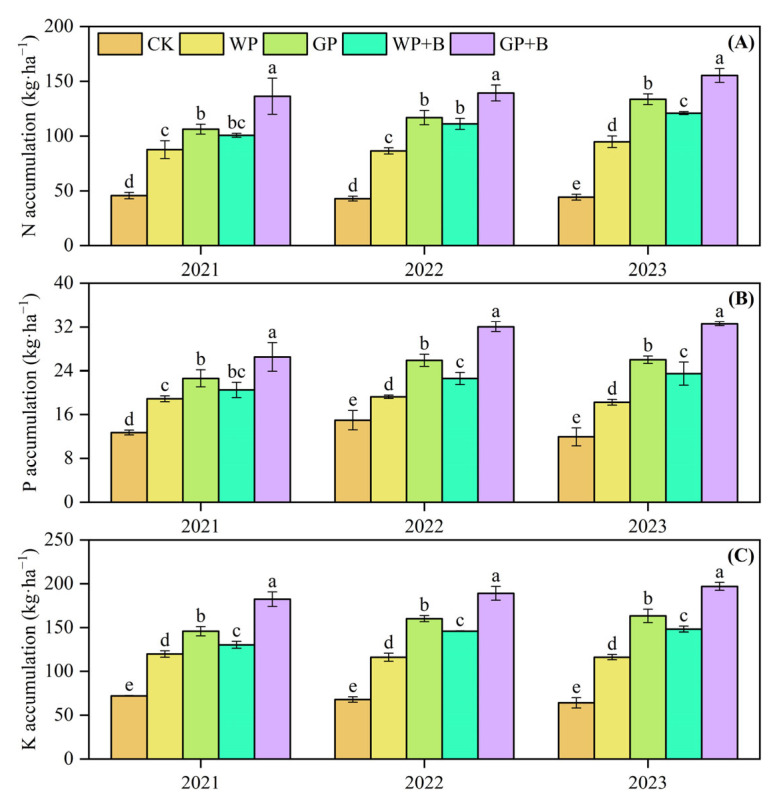
The effects of green manure return and biochar application on N (**A**), P (**B**), and K (**C**) nutrient accumulation in 2021–2023. Different lowercase letters indicate significant differences among different treatments at *p* < 0.05 by the Duncan’s MRT method.

**Figure 4 plants-13-03387-f004:**
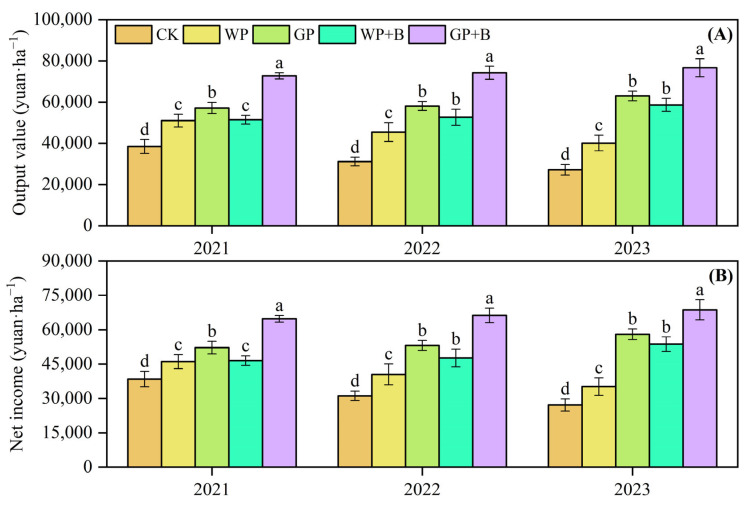
Effects of green manure return and biochar application on output value (**A**) and net income (**B**) of continuous cropping peppers in 2021–2023. Different lowercase letters indicate significant differences among different treatments at *p* < 0.05 by the Duncan’s MRT method.

**Table 1 plants-13-03387-t001:** Effects of green manure return and biochar application on nutritional quality of fresh pepper.

Year	Treatments	Free Amino Acid (g·kg^−1^)	Reducing Sugar (mg·kg^−1^)	VC (g·kg^−1^)	Nitrate (mg·kg^−1^)
2021	CK	3.64 ± 0.13 b	33.73 ± 1.73 d	0.82 ± 0.01 c	85.62 ± 3.58 d
	WP	3.72 ± 0.14 ab	36.18 ± 1.16 cd	0.84 ± 0.02 c	119.96 ± 1.60 a
	GP	3.76 ± 0.14 ab	41.16 ± 0.98 b	0.90 ± 0.03 b	107.13 ± 3.78 bc
	WP+B	3.75 ± 0.11 ab	38.94 ± 2.27 bc	0.89 ± 0.01 b	111.02 ± 3.45 b
	GP+B	3.93 ± 0.14 a	45.30 ± 2.86 a	0.95 ± 0.02 a	102.87 ± 3.79 c
2022	CK	3.66 ± 0.13 b	33.39 ± 2.22 c	0.67 ± 0.02 e	57.62 ± 0.67 d
	WP	3.73 ± 0.11 b	34.92 ± 0.35 c	0.74 ± 0.01 d	117.50 ± 7.40 a
	GP	4.07 ± 0.10 a	41.24 ± 1.94 b	0.94 ± 0.04 b	82.71 ± 0.95 b
	WP+B	4.05 ± 0.11 a	40.33 ± 0.47 b	0.84 ± 0.02 c	82.82 ± 0.33 b
	GP+B	4.22 ± 0.10 a	48.45 ± 2.29 a	1.07 ± 0.02 a	66.13 ± 2.16 c
2023	CK	3.66 ± 0.16 c	31.28 ± 1.70 c	0.52 ± 0.03 d	40.18 ± 1.40 c
	WP	3.76 ± 0.07 c	34.72 ± 2.49 c	0.63 ± 0.03 c	112.95 ± 5.74 a
	GP	4.53 ± 0.10 b	43.45 ± 2.22 b	1.03 ± 0.01 b	57.86 ± 2.01 b
	WP+B	4.38 ± 0.05 b	42.63 ± 2.14 b	1.00 ± 0.01 b	63.46 ± 2.03 b
	GP+B	5.25 ± 0.13 a	48.58 ± 1.99 a	1.09 ± 0.02 a	42.29 ± 3.21 c

Note: Different lowercase letters in the same column indicate significant differences among different treatments at *p* < 0.05 level by the Duncan’s MRT method.

**Table 2 plants-13-03387-t002:** Effects of green manure return and biochar application on fertilizer utilization of continues cropping peppers.

Year	Treatments	AE (kg·kg^−1^)			RE (%)		
		AE_N_	AE_P_	AE_K_	RE_N_	RE_P_	RE_K_
2021	CK	—	—	—	—	—	—
	WP	2.64 ± 0.83 c	9.97 ± 3.13 c	3.09 ± 0.97 c	17.68 ± 4.25 c	9.81 ± 1.43 c	23.54 ± 1.59 e
	GP	3.93 ± 1.00 c	14.86 ± 3.78 c	4.60 ± 1.17 c	25.53 ± 1.99 b	15.72 ± 3.18 abc	36.32 ± 2.52 c
	WP+B	2.73 ± 0.83 c	10.32 ± 3.14 c	3.19 ± 0.97 c	23.19 ± 1.45 bc	12.37 ± 2.63 bc	28.68 ± 1.86 d
	GP+B	7.21 ± 0.58 a	27.23 ± 2.18 a	8.43 ± 0.67 a	38.17 ± 5.72 a	21.94 ± 4.67 a	54.30 ± 4.04 a
2022	CK	—	—	—	—	—	—
	WP	3.00 ± 1.10 d	11.36 ± 4.17 d	3.51 ± 1.29 d	18.32 ± 1.70 c	6.80 ± 2.99 e	23.80 ± 2.92 d
	GP	5.67 ± 0.28 c	21.41 ± 1.07 c	6.63 ± 0.33 c	31.17 ± 3.67 b	17.37 ± 1.99 c	45.47 ± 2.72 b
	WP+B	4.52 ± 1.20 cd	17.07 ± 4.53 cd	5.28 ± 1.40 cd	28.72 ± 1.24 b	12.11 ± 1.12 d	38.39 ± 1.74 c
	GP+B	9.06 ± 0.93 a	34.25 ± 3.53 a	10.60 ± 1.09 a	40.60 ± 4.00 a	27.13 ± 1.55 a	59.70 ± 4.34 a
2023	CK	—	—	—	—	—	—
	WP	2.72 ± 0.78 d	10.28 ± 2.97 d	3.18 ± 0.92 d	21.29 ± 1.09 d	10.06 ± 2.37 e	25.66 ± 4.19 e
	GP	7.53 ± 1.04 bc	28.47 ± 3.93 bc	8.81 ± 1.22 bc	37.63 ± 1.75 b	22.37 ± 2.09 c	48.83 ± 2.97 c
	WP+B	6.62 ± 0.31 c	25.02 ± 1.17 c	7.75 ± 0.36 c	32.30 ± 0.75 c	18.35 ± 2.27 d	41.39 ± 1.26 d
	GP+B	10.41 ± 1.48 a	39.35 ± 5.58 a	12.18 ± 1.73 a	46.74 ± 1.98 a	32.84 ± 2.13 a	65.43 ± 4.69 a

Note: AE_N_, AE_P_, and AE_K_ stand for agronomic efficiency of N, P, and K. RE_N_, RE_P_, and RE_K_ stand for recovery efficiency of N, P, and K. Different lowercase letters in the same column indicate significant differences among different treatments at *p* < 0.05 level by the Duncan’s MRT method.

**Table 3 plants-13-03387-t003:** Green manure seeding, fertilizer, and biochar application rates for different treatments.

Treatments	Winter (Green Manure)	Summer (Pepper)		
	Seeding Rate (kg·ha^−1^)	CF1 (kg·ha^−1^)	CF2 (kg·ha^−1^)	Biochar (kg·ha^−1^)
CK	—	—	—	—
WP	—	—	720	—
GP	75	720	720	—
WP+B	—	720	720	1500
GP+B	75	720	720	1500

## Data Availability

The data supporting this study are available upon request.

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
