# Peer review of "Green Manure Rotation Combined with Biochar Application Improves Yield and Economic Stability of Continuous Cropping of Peppers in Southwest China"

_plants, 2024, doi:10.3390/plants13233387_

Round 1

Reviewer 1 Report

Comments and Suggestions for Authors

Please find the review in the file attached

Author Response

Thank you very much for taking your time to review this manuscript and give us another chance to revise the manuscript. We will cherish the opportunity of this revision. ​Your comments make this paper completer and more outstanding! I really appreciate all your comments and suggestions! Please find my itemized responses in below and my revisions/corrections in the re-submitted files. Thanks again.

  1. Novelty of the presented results is not stated clearly. The aim of the studies could be defined in a broader perspective. I mean, that such long term studies will be more valuable if the effect is evaluated not only for the small region in China, but if more general conclusion is drawn on the basis of the obtained results.

Thank you very much for your suggestion. We have made revisions to the entire manuscript in the hope of improving its quality.

  1. Whether biochar obtained from different feedstock or in different conditions will have the same effect on the soil and crops? Please discuss it in the article.

Thank you very much for your suggestion. We have added a discussion about biochar in our discussion.

  1. Line 108: WP+B experiment is without the use of the green manure according to the Materials and Methods section.

Thank you very much for your suggestion. We have further revised the explanations for each treatment in the materials and methods section.

  1. Line 134: What is VC? Abbreviations should be explained when they appear for the first time in the text.

Thank you very much for your suggestion. We have added an explanation for VC.

  1. Lines 149-150: It is not clear in this paragraph in what material the NPK content was measured. Was it plant, soil, or fruit?

Thank you very much for your suggestion. We have added an explanation regarding the NPK content.

  1. Table 2 is poorly readable. Please increase its width, decrease font size or make other changes in order to make it more readable.

Thank you very much for your suggestion. We have made modifications and improvements to Table 2.

  1. Lines 190-198: Information provided in the first part of the discussion was already given in the introduction and I see no point in repeating it here. So long introduction to the discussion is not necessary.

Thank you very much for your suggestion. We have removed duplicate discussions.

  1. Line 215: It should be ‘reducing sugars’.

Thank you very much for your suggestion. We have corrected the error.

  1. Lines 229-239: Since the presented studies do not cover topic of the microorganisms in the soil, such broad description about the effect of the continuous cropping on the microorganism community in the soil is not necessary in the discussion section and is out of the scope of the presented studies.

Thank you very much for your suggestion. We have removed irrelevant discussions.

  1. Line 242: Yield is not a factor that says anything about fruit quality, only about the total mass of fruits (in this case). Please correct it.

Thank you very much for your suggestion. We have corrected the error.

  1. Lines 271-275: Biochar can also release harmful compounds to the soil (see e.g., DOI: 10.3390/su131810362). Please discuss it here.

Thank you very much for your suggestion. We have added a discussion about biochar.

  1. Line 276: According to the presented results effect of biochar supplementation wasn't negative, it was only inferior to the green manure effect.

Thank you very much for your suggestion. We have corrected the error.

  1. Lines 282-286: It was not demonstrated in the presented studies, since neither soil microorganisms, nor plant roots were studied in this experiment. This may be explanation of the obtained results (yield, content of nutrients and other studied factors), but not a proof of them.

Thank you very much for your suggestion. We have revised and improved the relevant discussions.

  1. Lines 291-294: Conclusion and future plans should be transferred to the conclusion section.

Thank you very much for your suggestion. We have revised and improved the conclusion section.

  1. Lines 304-308: This paragraph should be transferred to the results section, since it describes results of the soil analysis, which was performed in the studies and which is described in the section 4.4.

Thank you very much for your suggestion. The soil analysis data were only measured as basic data and are not part of the results of this study. Therefore, we only included them in the Materials and Methods section.

  1. Line 312: Why this genus of plant was used as a green manure? Please include a short answer in the manuscript.

Thank you very much for your suggestion. We have added an introduction to the green manure used in the experiment.

  1. Line 313: Please include information on fertilizer producer.

Thank you very much for your suggestion. We have added information on fertilizer manufacturers and other related details.

  1. Line 314-315: Description of the used biochar is very scarce. Is it commercial biochar? If so, who was supplier? What was characteristic of the used biochar? Was it in powder form or granular? Biochar characteristic is important for its effectiveness, so this data should be provided in the paper.

Thank you very much for your suggestion. We have added an introduction to the biochar used in the experiment.

  1. Line 321: Winter fallow and pepper rotation was used also on CK fields, wasn't it? It seems like the only difference between CK and WP is the use of mineral fertilizers CF1 and CF2. The description is a little misleading. Please make it clearer.

Thank you very much for your suggestion. We have further revised the explanations for each treatment in the materials and methods section.

  1. 20.Line 345: Abbreviations should be explained when they appear for the first time in the article

Thank you very much for your suggestion. We have provided corresponding explanations.

Reviewer 2 Report

Comments and Suggestions for Authors

The manuscript entitled "Green manure rotation combined with biochar application improves yield and economic stability of continuous cropping of peppers in Southwest China’’ investigated the effects of green manure rotation and biochar application on the continuous cropping of peppers. The experiment consisted of five treatments: CK (no green manure and no biochar), WP (winter fallow and pepper rotation), GP (green manure and pepper rotation), WP + B (winter fallow and pepper rotation with biochar applied during the pepper season), and GP + B (green manure and pepper rotation with biochar applied during the pepper season). Overall, the paper is well written, but some comments should be modified. In the text, some text formatting mistakes should be modified, for example, lines 103, 148, and …

Please see more comments below:

·         In the Abstract section, there is no information about the application rate of biochar and green manure and should be added. In the results section, there isn’t any comparison between treatments with the percentage. The authors should give some information regarding increased or decreased changes between treatments. And, finally, keywords should differ from the title!

·         In the introduction section, add previous research focus on green manure and biochar alone and combined together on different crops, compare them with your work, and explain what is the gap of knowledge in these fields and how your work can fill this gap. Here is a recently published paper and you can use it here: https://doi.org/10.1016/j.jclepro.2024.143772

·         There is no information about the research hypothesis at the end of the abstract section or at the end of the introduction section.

·         Add error bars and significant letters for Figures 2 (A-D).

·         There is no information about VC (vitamin C) calculation or measurements in the material and method section.

·         The results of the section 2.4. nutrient accumulation in continuous cropping peppers is not enough. I recommend rewriting this section.

·         The main problem in the discussion section is related to the comparison of the current with other research, but in the current research, there is no direct comparison. In some parts, the authors refer to previous research (Lines 229-231) but it is general.

·         Add the latitude and longitude of the research area in the section4.1. site description.

·         Add the soil characteristics analysis in the material and method section.

·         Add the study limitations and suggestions for future research in the conclusion section.

·         In reference numbers: 5, 12, 13, 20, 31, 42, 43, 45,47, 51, 69, 71, and 73 are related to the author's self-citation, I would suggest that remove or replace them. 

Author Response

Thank you very much for taking your time to review this manuscript and give us another chance to revise the manuscript. We will cherish the opportunity of this revision. ​Your comments make this paper completer and more outstanding! I really appreciate all your comments and suggestions! Please find my itemized responses in below and my revisions/corrections in the re-submitted files. Thanks again.

  1. In the Abstract section, there is no information about the application rate of biochar and green manure and should be added. In the results section, there isn’t any comparison between treatments with the percentage. The authors should give some information regarding increased or decreased changes between treatments. And, finally, keywords should differ from the title!

Thank you very much for your suggestion. We have further revised and improved the abstract section.

  1. In the introduction section, add previous research focus on green manure and biochar alone and combined together on different crops, compare them with your work, and explain what is the gap of knowledge in these fields and how your work can fill this gap. Here is a recently published paper and you can use it here: https://doi.org/10.1016/j.jclepro.2024.143772

Thank you very much for your suggestion. We have added some research progress on green manure or biochar in the introduction.

  1. There is no information about the research hypothesis at the end of the abstract section or at the end of the introduction section.

Thank you very much for your suggestion. We have added research hypotheses at the end of the introduction.

  1. Add error bars and significant letters for Figures 2 (A-D).

Thank you very much for your suggestion. We have modified Figure 2 and improved the experimental results.

  1. There is no information about VC (vitamin C) calculation or measurements in the material and method section.

Thank you very much for your suggestion. We have supplemented the measurement methods for indicators such as VC.

  1. The results of the section 2.4. nutrient accumulation in continuous cropping peppers is not enough. I recommend rewriting this section.

Thank you very much for your suggestion. We have added relevant descriptions.

  1. The main problem in the discussion section is related to the comparison of the current with other research, but in the current research, there is no direct comparison. In some parts, the authors refer to previous research (Lines 229-231) but it is general.

Thank you very much for your suggestion. We have revised and improved the discussion section, please refer to the revised draft for details.

  1. Add the latitude and longitude of the research area in the section4.1. site description.

Thank you very much for your suggestion. We have provided a detailed description of the research site.

  1. Add the soil characteristics analysis in the material and method section.

Thank you very much for your suggestion. We have provided a detailed description of the soil type.

  1. Add the study limitations and suggestions for future research in the conclusion section.

Thank you very much for your suggestion. We have added the limitations of the research and future recommendations.

  1. In reference numbers: 5, 12, 13, 20, 31, 42, 43, 45,47, 51, 69, 71, and 73 are related to the author's self-citation, I would suggest that remove or replace them.

Thank you very much for your suggestion. We have made modifications and replacements to some of the references.

Round 2

Reviewer 2 Report

Comments and Suggestions for Authors

.

Comments on the Quality of English Language

Minor English editing is required. 

Author Response

Thank you very much for your suggestion. We have improved our English language proficiency.